# *Kdm6a* deficiency restricted to mouse hematopoietic cells causes an age- and sex-dependent myelodysplastic syndrome-like phenotype

Ling Tian[1][☯], Monique Chavez[1][☯], Gue Su Chang[2], Nichole M. Helton[1], Casey D. S. Katerndahl[1], Christopher A. Miller[1,2], Lukas D. Wartman[1]*

**1** Division of Oncology, Department of Internal Medicine, Washington University School of Medicine, St Louis, MO, United States of America, **2** McDonnell Genome Institute, Washington University School of Medicine, St. Louis, MO, United States of America

☯ These authors contributed equally to this work.
* lwartman@wustl.edu

**Data Availability Statement:** Sequence data is available at NCBI under BioProject PRJNA735514.

## Abstract

*Kdm6a/Utx*, a gene on the X chromosome, encodes a histone H3K27me3 demethylase that has an orthologue on the Y chromosome (*Uty*) (Zheng et al. 2018). We previously identified inactivating mutations of *Kdm6a* in approximately 50% of mouse acute promyelocytic leukemia samples; however, somatic mutations of *KDM6A* are more rare in human AML samples, ranging in frequency from 2–15% in different series of patients, where their role in pathogenesis is not yet clear. In this study, we show that female *Kdm6a^flox/flox* mice (with allele inactivation initiated by *Vav1*-Cre in hematopoietic stem and progenitor cells (HSPCs) have a sex-specific phenotype that emerges with aging, with features resembling a myelodysplastic syndrome (MDS). Female *Kdm6a*-knockout (KO) mice have an age-dependent expansion of their HSPCs with aberrant self-renewal, but they did not differentiate normally into downstream progeny. These mice became mildly anemic and thrombocytopenic, but did not develop overt leukemia, or die from these cytopenias. ChIP-seq and ATAC-seq studies showed only minor changes in H3K27me3, H3K27ac, H3K4me, H3K4me3 and chromatin accessibility between *Kdm6a*-WT and *Kdm6a*-KO mice. Utilizing scRNA-seq, *Kdm6a* loss was linked to the transcriptional repression of genes that mediate hematopoietic cell fate determination. These data demonstrate that *Kdm6a* plays an important role in normal hematopoiesis, and that its inactivation may contribute to AML pathogenesis.

## Introduction

The human *KDM6A* gene is located on chromosome Xp11.2 and contains a catalytically-active Jumonji C demethylase domain with activity directed towards the di- and tri-methylated lysine 27 of histone H3 (H3K27) [2]. H3K27 trimethylation is associated with gene silencing, whereas the trimethylation of H3K4 is associated with gene transcription [3]. *KDM6A* has established

**Funding:** This work was supported by institutional funds from the Division of Oncology, the Barnes Jewish Hospital Foundation, and the McDonnell Genome Institute. It was also supported by the following National Institutes of Health grants: K08 CA166229 (L.D.W.), P01 CA101937 (T.J. Ley, PI, C.A.M, Core C leader), and R50 CA211782 (C.A. M.). Flow cytometry was performed in the Siteman Cancer Center Flow Cytometry Core, which is supported in part by National Institutes of Health, National Cancer Institute Cancer Center support grant P30 CA91842. The funders had no role in study design, data collection and analysis, decision to publish, or preparation of the manuscript.

**Competing interests:** The authors have declared that no competing interests exist.

roles in development and stem cell differentiation [4–10]. Homozygous constitutive *Kdm6a*-null female mice die before embryonic day 13.5. About 25% of *Kdm6a*-null hemizygous male mice survive, but are runted and have a reduced lifespan. *Kdm6a*-null mice apparently die from cardiac defects, but they also have neural tube defects and anemia/yolk sac abnormalities [6–8, 11]. KDM6A associates with mixed-lineage leukemia (MLL) 2/3-containing complexes, which have H3K4 methyltransferase activity [12]; inactivation of *KDM6A* could therefore lead to perturbations in H3K4 methylation. Recent work has demonstrated that KDM6A interacts with KMT2D and p300 (the H3K27 acetyltransferase) in a demethylase-independent manner, changing the histone marks of enhancers [13]. Therefore, alterations in *KDM6A* expression could lead to changes in H3K27 acetylation as well [14].

We previously identified recurrent loss-of-function somatic mutations and deletions of *Kdm6a* in a mouse model of acute promyelocytic leukemia [15]. Other groups have identified inactivating *KDM6A* mutations in a wide variety of tumor types [16, 17]. In the acute myelocytic leukemia (AML) TCGA dataset, *KDM6A* somatic mutations were identified in 4 of 200 AML patients [18]. Another large sequencing study reported a lower incidence of *KDM6A* driver mutations in AML (10 of 1540 cases), whereas a more recent study used a different gene panel and reported an incidence of 27% (32 of 119 cases) [19, 20]. *KDM6A* mutations are enriched in cases of core-binding factor AML initiated by the *RUNX1-RUNX1T1* fusion gene, where 6–7% of these patients also have mutations in *KDM6A* [21, 22].

*KDM6A* mutations also occur in acute lymphoblastic leukemia (ALL), and in chronic myelomonocytic leukemia (CMML) [23–28]. The somatic inactivation of *KDM6A* has been suggested to cause the dysregulation of HOX, NOTCH, and/or RB signaling pathways, which could be relevant for leukemogenesis [4, 29–32]. In T-ALL, there is evidence to suggest that inactivating mutations in *KDM6A* act via a demethylase-dependent mechanism [28]. However, it remains unclear how the loss of *KDM6A* demethylase activity contributes to T-ALL, or how *KDM6A* mutations contribute to AML pathogenesis.

In this study, we examined the role of *Kdm6a* in normal hematopoiesis by inactivating *Kdm6a* in the hematopoietic stem and progenitor cells (HSPCs) of adult mice of both sexes, and defined the epigenetic consequences as a function of age. We also compare the findings in these mice to that of other previously published mouse models of Kdm6a deficiency [1, 33–36], to highlight similarities and differences in phenotypes that may be due to how the gene is inactivated in adult mice (i.e. Cre transgene strategies) or other factors. This study represents the only described model of inactivation that is hematopoietic-specific, and it has some unique features that help to define the cell autonomous features of the phenotype.

## Methods

### Mouse generation

All mice experiments were performed in accordance with institutional guidelines and current NIH policies and were approved by the Animal Studies Committee at Washington University. Details for the C57BL/6J mouse strains and husbandry are provided in the supplementary methods.

## Serial replating assay

The serial replating assay was based on previous experiments by Cole et al. [37], and additional paper-specific details are found in supplementary methods.

### Transplantation assays

BM and spleen cells were harvested and used for transplantation. Additional information about the transplantation assay can be found in supplementary methods.

### Statistics

Unpaired two-tailed Student's *t*-test and log-rank test were calculated using Prism 7 software. *P* values < 0.05 were considered statistically significant. All data presented the mean ± SD unless otherwise indicated.

### Bioinformatics and associated statistical analysis

The bioinformatics and associated statistical analysis are detailed in the supplementary methods. For all experiments, additional details can be found in supplementary methods (S1 Methods). Sequence data is available at NCBI under BioProject PRJNA735514.

## Results

### Inactivation of *Kdm6a* causes age- and gender-dependent effects on normal hematopoiesis

We generated a conditional knockout mouse using ES cells containing a targeted *Kdm6a*<sup>flox</sup> allele obtained from the EUCOMM resource [38]. We crossed *Kdm6A*<sup>flox/flox</sup> mice with *Vav1-Cre* transgenic mice to inactivate *Kdm6a* in HSPCs during mid-gestation (**Fig 1A**) [39]. All mice were continuously housed in a Specific Pathogen Free (SPF) barrier facility. Our study included female homozygous (KO-F), female heterozygous (Het-F), male hemizygous (KO-M), and appropriate age-matched littermate control (WT-F or WT-M) mice. We verified targeting of the *Kdm6a* conditional KO allele via whole genome sequencing of DNA isolated from bulk bone marrow (BM) of a young KO-M mouse. We did not detect Kdm6a protein expression in BM cells isolated from KO-F or KO-M mice expressing *Vav1*-Cre (**Fig 1B**).

 We characterized hematopoiesis in young (<12 weeks) and aged (50- to 55-week-old) *Kdm6a* conditional KO mice and controls. Complete blood counts (CBCs) revealed mild thrombocytopenia in young and aged KO-F mice. Aged KO-F mice developed mild macrocytic anemia (**Fig 1C**). There were no significant differences in the blood counts of young or aged Het-F or KO-M mice (**S1A and S1B Fig**). Young KO-F mice had mild splenomegaly, which grew more pronounced with age (**Fig 1D**). We observed mild splenomegaly in young KO-M mice, but not in aged KO-M or Het-F mice (**S1C Fig**). Cytospin preparations of aged KO-F BM demonstrated increased precursors with binucleated and dysplastic erythroid precursors, and atypical megakaryocytes, both consistent with myelodysplasia using the Bethesda Criteria (**Fig 1E**) [40]. The bone marrow of young and aged KO-F mice showed moderately increased cellularity with increased numbers of myeloid cells and decreased numbers of erythroid cells. The spleens from young KO-F mice had a mild increase in extramedullary erythropoiesis and granulopoiesis (**S1D Fig**).

 We analyzed frequencies of myeloid, B-, T-, NK, and erythroid precursor cells in BM, spleen, and peripheral blood (PB). KO-F mice had an increased number of myeloid cells in the BM and spleen that became more pronounced with age. The frequency of myeloid cells in PB was increased in aged mice. We found a decreased number of erythroid precursors in BM of young and aged KO-F mice, which was compensated by an increase of these cells in the spleen (**Fig 1F and 1G**). Young and aged Het-F and KO-M mice did not have significant differences in these populations, with the exception of a mild B-cell reduction in spleens of young KO-M mice (**S2A and S2B Fig**).

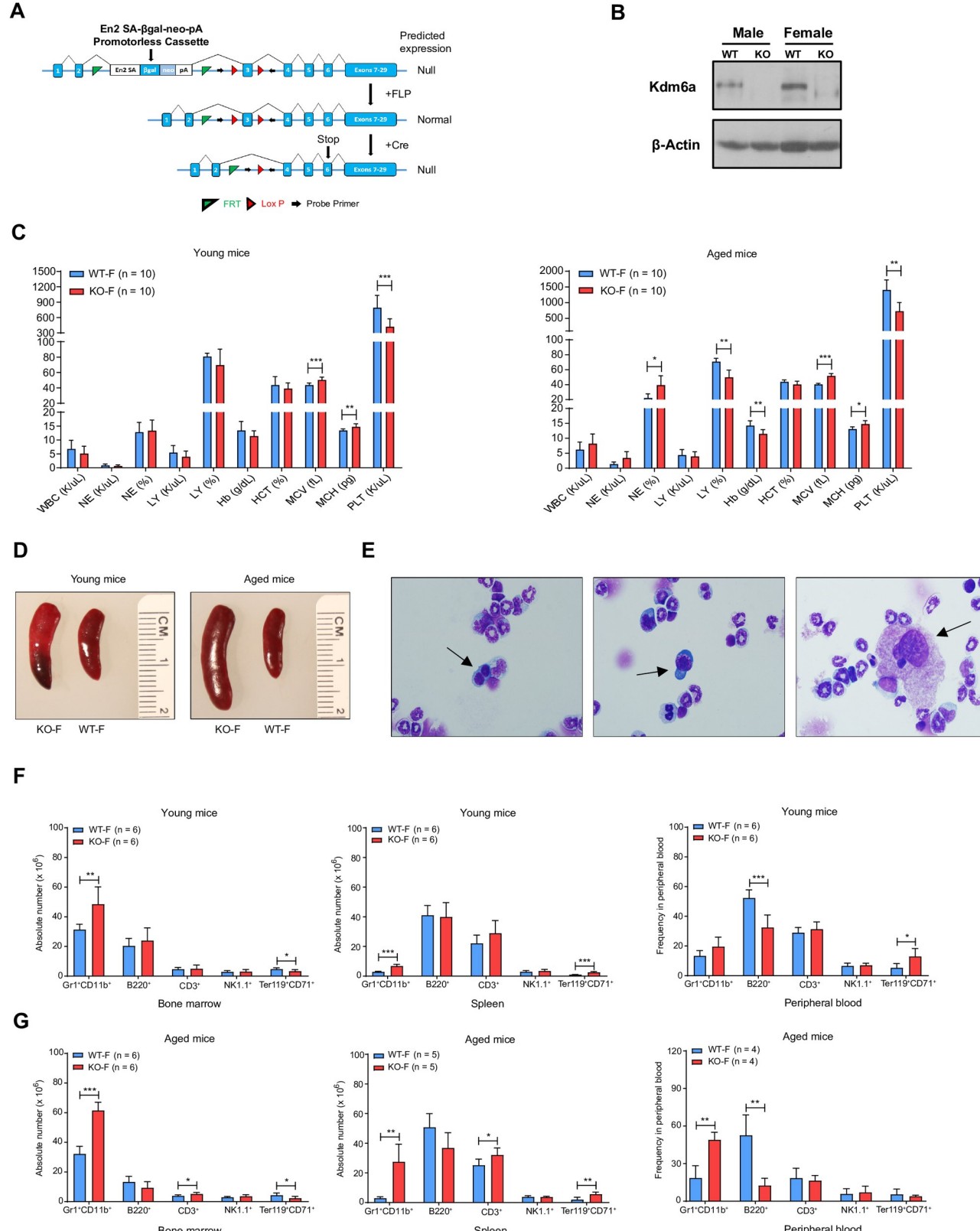

**Fig 1. The inactivation of *Kdm6a* in mice had age- and gender-dependent effects on normal hematopoiesis.** (A) The targeting strategy of the *Kdm6a* conditional knockout allele (Cre-mediated recombination leads to a frameshift mutation that creates a stop codon at exon 6). (B) Western blot

analysis of protein lysate from whole bone marrow (BM) cells showed efficient knockout of *Kdm6a* in female homozygous (KO-F) and male hemizygous (KO-M) mice (6–8 weeks old). Data for littermate control mice (WT-F or WT-M) is also presented. (C) Complete blood counts (CBC) identified mild thrombocytopenia in both young (422.3 K/uL in KO-F mice vs. 793.8 K/uL in WT-F mice, p = 0.0006) and aged (728.9 K/uL in KO-F mice vs. 1405 K/uL in WT-F mice, p = 0.0028) mice. Aged KO-F mice developed a mild macrocytic anemia, which was characterized by decreased hemoglobin (Hb: 11.5 g/dL in KO-F mice vs. 14.28 g/dL in WT-F mice, p = 0.0137) with an increased mean corpuscular volume (MCV: 51 fL in KO-F mice vs. 40.55 fL in WT-F mice, p = 0.0001) and mean corpuscular hemoglobin (MCH: 15.15 pg in KO-F mice vs. 12.85 pg in WT-F mice, p = 0.0004). (D) Representative images of spleen showed mild splenomegaly in young KO-F mice compared with WT-F mice. These findings were more pronounced in aged mice. (E) Histopathologic examination of BM with Wright Giemsa staining. Images are representative of 3 independent cytospin preparations. Black arrows indicate examples of myelodysplasia. (F) Cell lineage absolute numbers for BM and spleen and frequency for peripheral blood (PB) from young, KO-F mice and WT-F mice were determined by flow cytometry for myeloid cells, B cells, T cells, NK cells and erythroid precursors. There was a significant increase in number of Gr1$^+$CD11b$^+$ myeloid cells in BM and spleen in KO-F mice compared to control. In KO-F mice there was also a significant decrease in Ter119$^+$CD71$^+$ erythroid precursors in BM and significant increase Ter119$^+$CD71$^+$ erythroid precursors in spleen and PB. (G) Cell lineage absolute numbers for BM and spleen and frequency for PB from aged, KO-F mice and WT-F mice were determined by flow cytometry for myeloid cells, B cells, T cells, NK cells and erythroid precursors. There was a significant increase in the number of Gr1$^+$CD11b$^+$ myeloid cells in BM, spleen and PB of KO-F mice compared to control. There was a significant decrease in B220$^+$ B cells in the PB of KO-F mice. There remained a significant decrease in Ter119$^+$CD71$^+$ erythroid precursors in the BM and significant increase in this cell population in spleen and PB in KO-F mice. Error bars represented mean ± s.d. *p<0.05, **p<0.01, ***p<0.001 by unpaired two-tailed Student's *t*-test.

To determine whether loss of *Kdm6a* in HSPCs influenced self-renewal, we performed serial replating assays using whole BM cells isolated from young and aged mice (**Fig 2A**). BM cells from young KO-F and KO-M mice displayed a myeloid replating phenotype indicative of aberrant self-renewal (**Fig 2B and 2C**). This phenotype was also present in cells derived from aged mice. Het-F mice acquired an abnormal replating phenotype with age (**Fig 2D**). Flow cytometry analysis of BM cells isolated from KO-F mice after two cycles of replating showed a significant increase in the proportion of mature myeloid cells, compared to control mice in both young and aged mice (**Fig 2E**).

To investigate the effects of *Kdm6a* inactivation on primitive hematopoietic cell populations, we analyzed progenitor compartments in BM from young or aged *Kdm6a* conditional KO mice. We detected an increase in the frequency of HSPCs, as defined by SLAM flow markers in aged KO-F mice (**Fig 3A**). There was a significant decrease of the LSK cell population in young KO-F mice, with a concomitant decrease in multipotent progenitors (MPPs), which remained significant in aged KO-F mice (**Fig 3B**). In myeloid progenitors, we observed a decrease in megakaryocyte-erythrocyte progenitors (MEPs) and an increase in granulocyte-macrophage progenitors (GMPs). In BM from aged KO-F mice, the MEP compartment remained significantly smaller (**Fig 3C**). The phenotypes described above were specific to KO-F mice and were not present in KO-M or Het-F mice (**S3A–S3D Fig**), which indicated that inactivation of *Kdm6a* had age- and sex-dependent effects on primitive hematopoietic cell populations.

Because loss of *Kdm6a* in KO-F mice was associated with a decrease in the LSK compartment, we characterized cell cycle kinetics and measured apoptosis of that population using flow cytometry. The LSK population in young KO-F mice had an increased fraction in the G0/G1 phase of the cell cycle, and a greater proportion of LSK cells with late apoptotic features (**Fig 3D**).

### *Kdm6a* conditional KO BM cells have decreased repopulating potential

To evaluate the effects of *Kdm6a* inactivation on HSPC self-renewal and differentiation *in vivo*, we performed competitive transplantation experiments (**Fig 4A**). We noted a significant decrease in peripheral blood chimerism in the KO-F donor cell cohort, 4 weeks after transplant. Donor cells from Het-F and KO-M mice had a significant competitive disadvantage compared to WT competitor donor cells and did not have cell migration or engraftment problems. However, there may have been cell migration and subtle engraftment problems at an early timepoint after transplantation that our current assay and measurements did not account for (**Fig 4B**). 36 weeks post-transplantation, we assessed donor cell chimerism in BM HSPC populations and mature cell lineages from all cohorts. There was a selective preservation of

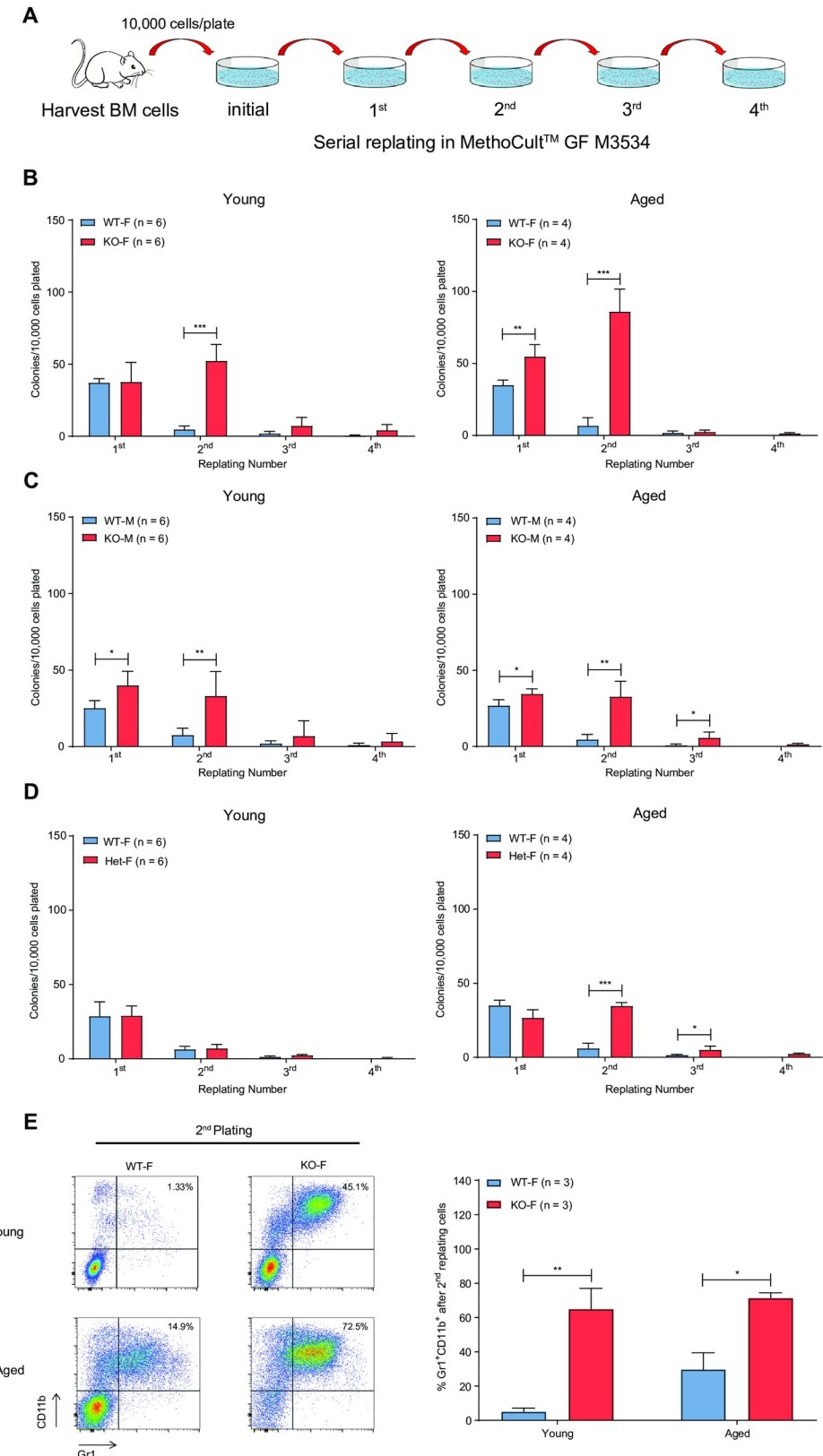

**Fig 2. *Kdm6a* conditional KO induced aberrant myeloid self-renewal.** (A) Experimental schema of the serial replating assay. Whole BM cells were harvested from young and aged *Kdm6a* conditional KO mice and appropriate

littermate controls. 10,000 cells were serially replated in MethoCult™GF M3534 for 4 weeks. (B-D) Quantification of colony numbers showed a myeloid replating phenotype in KO-F mice (10.40-fold increase in colonies from KO-F mice compared to those from WT-F mice after 2 rounds of replating, p = 0.0001) and KO-M mice (4.12-fold increase in colonies from KO-M mice compared to those from WT-M mice after 2 rounds of replating, p = 0.0001) indicative of aberrant self-renewal. This was present in both young and aged mice with increased colony numbers at week 2 in KO-F mice as compared to KO-M mice. Het-F mice only acquired this abnormal replating phenotype with age. (E) Representative flow cytometry results of BM cells isolated from KO-F mice after two cycles of replating showed a significant increase in Gr1$^+$CD11b$^+$ myeloid cells as compared to control mice (left panel). Quantification of the indicated flow cytometry results was significant in cells from both young (12.5-fold increase in KO-F mice compared to those from WT-F mice, p = 0.0083) and aged mice (2.4-fold increase in KO-F mice compared to those from WT-F mice, p = 0.0149, right panel). Error bars represented mean ± s.d. *p<0.05, **p<0.01, ***p<0.001 by unpaired two-tailed Student's $t$-test.

CD45.2$^+$ SLAM cells from KO-F donors, compared to a near complete loss of HSPC subpopulations and mature cell lineages. In Het-F and KO-M cohorts, the CD45.2$^+$ SLAM compartment was also preserved (**Fig 4C**, **S4A–S4C Fig**).

We tested the repopulating potential of *Kdm6a* KO donor cells from primary transplants by performing secondary transplantation using whole BM isolated at the end of the primary transplants. PB chimerism revealed a more pronounced competitive repopulation disadvantage in donor cells from all cohorts of *Kdm6a* KO mice (**Fig 4D**). Preservation of the SLAM compartment was sustained in secondary transplants using donor cells derived from KO-M mice at 18 weeks post-transplant (**Fig 4E**).

## The *Kdm6a*-null stroma promotes the survival of *Kdm6a*-null hematopoietic stem and progenitor cells

Since *Vav1*-Cre inactivates *Kdm6a$^{flox/flox}$* alleles in HSPCs (and with the possible exception of endothelial cells, not other BM stroma cells), and since other models of *Kdm6a* knockouts inactivated the gene in both compartments, we wished to define the effect of stromal *Kdm6a* inactivation for hematopoietic phenotypes [1, 33, 34, 40, 41]. We therefore performed non-competitive transplantation experiments using donor cells isolated from the BM of young *Kdm6A$^{flox/flox}$* x *Cre-ERT2$^{+/-}$* mice (with tamoxifen-inducible deletion of *Kdm6a*), which were transplanted into young lethally irradiated recipient mice (**Fig 5A and 5B**). Recipient cohorts were different from each other based on the presence or absence of *Kdm6a* conditional KO and *Cre-ERT2$^{+/-}$* alleles. KO-M x *Cre-ERT2$^{+/-}$* mice used as recipients would have genetic inactivation of *Kdm6a* in all tissues (including BM stroma) after tamoxifen administration. We measured *Kdm6a* floxing efficiency in BM by performing qPCR, and found that average floxing efficiency was 83.5% 6 weeks after administration of 9 doses of Tamoxifen by gavage (in cohorts 1 and 2). We detected no *Kdm6a* floxing in BM isolated from WT-F x *Cre-ERT2$^{+/-}$* or WT-M x *Cre-ERT2$^{+/-}$* cohorts not given Tamoxifen, as expected (cohort 4). We detected a floxing efficiency of 11.2% in cohort 3, which was probably due to the presence of *Cre-ERT2$^{+/-}$* in stromal cells of the recipient mice. The KO-F x *Cre-ERT2$^{+/-}$* donor and KO-M x *Cre-ERT2$^{+/-}$* recipient cohort (cohort 1) had a significant increase in the LSK and SLAM cell populations, compared to mice that expressed *Kdm6a* in the BM alone. A similar trend was seen with an increase in GMPs, and a decrease in MEPs (**Fig 5C and 5D**, **S5A and S5B Fig**). These results suggest that *Kdm6a* deficient stroma promotes the expansion of *Kdm6a*-null HSPCs.

## Loss of *Kdm6a* in hematopoietic cells does not lead to overt leukemia in *Vav1*-Cre expressing mice

To determine whether disruption of *Kdm6a* in HSPCs only (mediated by *Vav1*-Cre) might lead to hematopoietic malignancies, we performed an 18-month tumor watch. *Kdm6a*

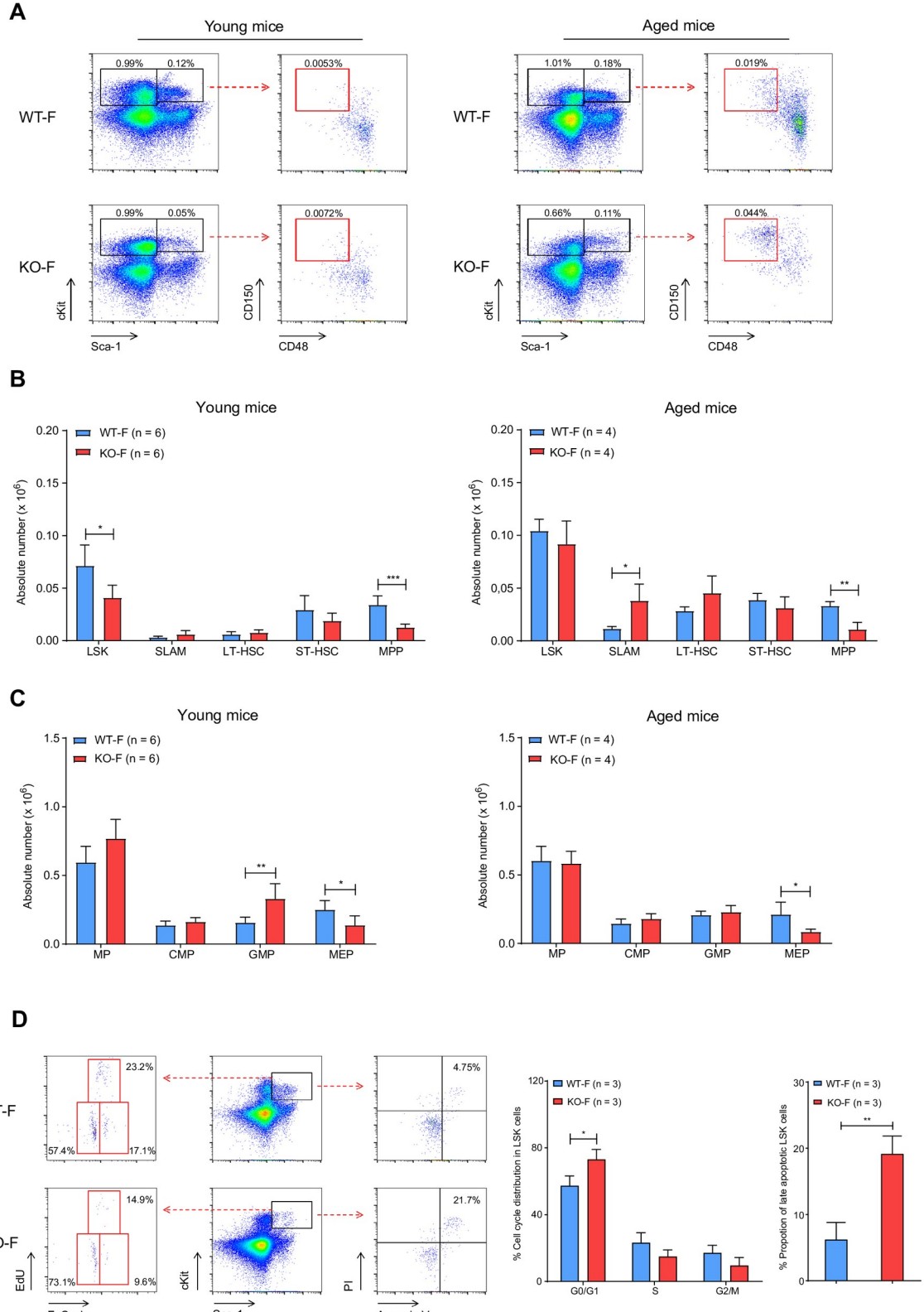

**Fig 3. Characterization of hematopoietic stem and progenitor cells from BM of young and aged KO-F mice vs. WT-F controls.**
(A) Representative flow cytometric plots of BM cells from young and aged KO-F mice and appropriate littermate controls. Profiles were gated on LSK (Lineage⁻Sca-1⁺cKit⁺) and MP (Lineage⁻Sca-1⁻ckit⁺) populations (black gates) and then the LSK compartment was

sub-gated for SLAM (Lineage$^-$Sca-1$^+$cKit$^+$Cd48$^-$Cd150$^+$, red gates) markers. (B) Quantification of the indicated stem cell and progenitor compartments showed a significant decrease in absolute number of cells in LSK (1.74-fold decrease in KO-F mice as compared to WT-F mice, p = 0.0115) and MPP (Lineage$^-$Sca-1$^+$cKit$^+$CD34$^+$Flt3$^+$, 2.71-fold decrease in young KO-F mice as compared to WT-F mice, p = 0.003) compartments in BM from young KO-F mice. BM from aged KO-F mice had a significant increase in absolute number of SLAM cells (3.32-fold increase in aged KO-F mice as compared to WT-F mice, p = 0.0150), and a concomitant decrease in the MPP compartment. (C) Quantification of indicated progenitor compartments showed a significant decrease in absolute number of cells in the MEP compartment (Lineage$^-$Sca-1$^-$cKit$^+$CD34$^-$FcgR$^{hi}$, 1.82-fold decrease in KO-F mice as compared to WT-F mice, p = 0.0087) and increase in GMP compartment (Lineage$^-$Sca-1$^-$cKit$^+$CD34$^{hi}$FcgR$^{hi}$, 2.11-fold increase in young KO-F mice as compared to WT-F mice, p = 0.0082) in BM from young KO-F mice. From aged KO-F mice, the MEP compartment was significantly decreased. (D) Cell cycle and apoptosis analysis of LSK cell populations from young KO-F mice and appropriate littermate controls by flow cytometry. Representative flow cytometric plots for apoptosis makers (Annexin V and PI) and cell cycle makers (EdU and FxCycle). Relative quantification of cell cycle distribution and the proportion of late apoptotic cells in the LSK population showed the decrease in the LSK compartment in young KO-F mice was associated with a cell cycle shift to the G0/G1 phase (1.26-fold increase in KO-F mice as compared to WT-F mice, p = 0.0305) and increased apoptosis (4.57-fold increase in KO-F mice as compared to WT-F mice, p = 0.0040). Error bars represented mean ± s.d. *p<0.05, **p<0.01, ***p<0.001 by unpaired two-tailed Student's *t*-test.

conditional KO mice did not develop spontaneous leukemias, or have an increased incidence of other cancers compared to control mice (S6A Fig). At the end of the tumor watch, all three cohorts of *Kdm6a* KO mice had mild splenomegaly, which was most pronounced in KO-F mice (S6B Fig). CBCs were essentially unchanged from mice that were 12 months of age (S6C Fig). Lineage flow cytometry showed infiltration of the spleens with mature myeloid cells and erythrocyte progenitors, with a concomitant decrease in B and T lymphocytes (S6D Fig).

To test for malignant potential of splenocytes isolated from 18-month-old mice, we did adoptive transfer experiments using 3 different donors (S6E Fig). Serial analysis of PB and BM revealed that donor-derived KO-F splenocytes failed to differentiate into mature cell lineages or give rise to leukemia. However, we detected an increased contribution to primitive hematopoietic cell populations from donor-derived cells after long-term observation (S6F Fig).

### Epigenetic analyses of *Kdm6a*-WT and *Kdm6a*-KO hematopoietic cells

To understand the epigenetic phenotypes of the hematopoietic cells of *Kdm6a*-null mice, we used unfractionated BM to perform single-cell RNA-sequencing (scRNA-seq), bulk RNA-sequencing (RNA-seq), chromatin immunoprecipitation with sequencing (ChIP-seq), and an assay for transposase-accessible chromatin using sequencing (ATAC-seq). We performed an ELISA-like profiling assay to assess for differences in total histone H3, and 21 different histone H3 modifications between young, KO-F mice vs. controls. An analogous assay for total histone H4 and 10 histone H4 modifications was done using the same samples. We detected no significant differences in the modifications tested. We performed immunoblotting using an anti-H3K27ac Ab, and detected significantly less H3K27ac, using unfractionated BM from young, KO-F mice vs controls (S7A–S7C Fig). Genome-wide changes in histone modifications were not apparent between KO-F and WT-F samples at any position of histone marks or ATAC-seq. We found slightly greater peak occupancy for KO-F samples vs. WT-F samples of H3K4me peaks using the *Kdm6a*-null peak calls (S8A–S8E Fig).

Our data included the epigenetic structures of 18,624 transcriptionally-engaged promoters of annotated genes, defined by significant enrichment of H3K4me3 peaks defined by ChIP-seq data. When plotted using H3K4me3 peaks from a combination of both genotypes using hierarchical clustering or peak density plots, we did not find large scale differences between KO-F and WT-F samples. DNA accessibility upstream from active promoters was also apparent, as shown by the distribution of ATAC-seq reads around the transcription start sites (TSS) (S7D Fig).

We also evaluated histone modifications and DNA accessibility at enhancers. First, we defined enhancer-like regions and candidate active enhancers using defined epigenetic

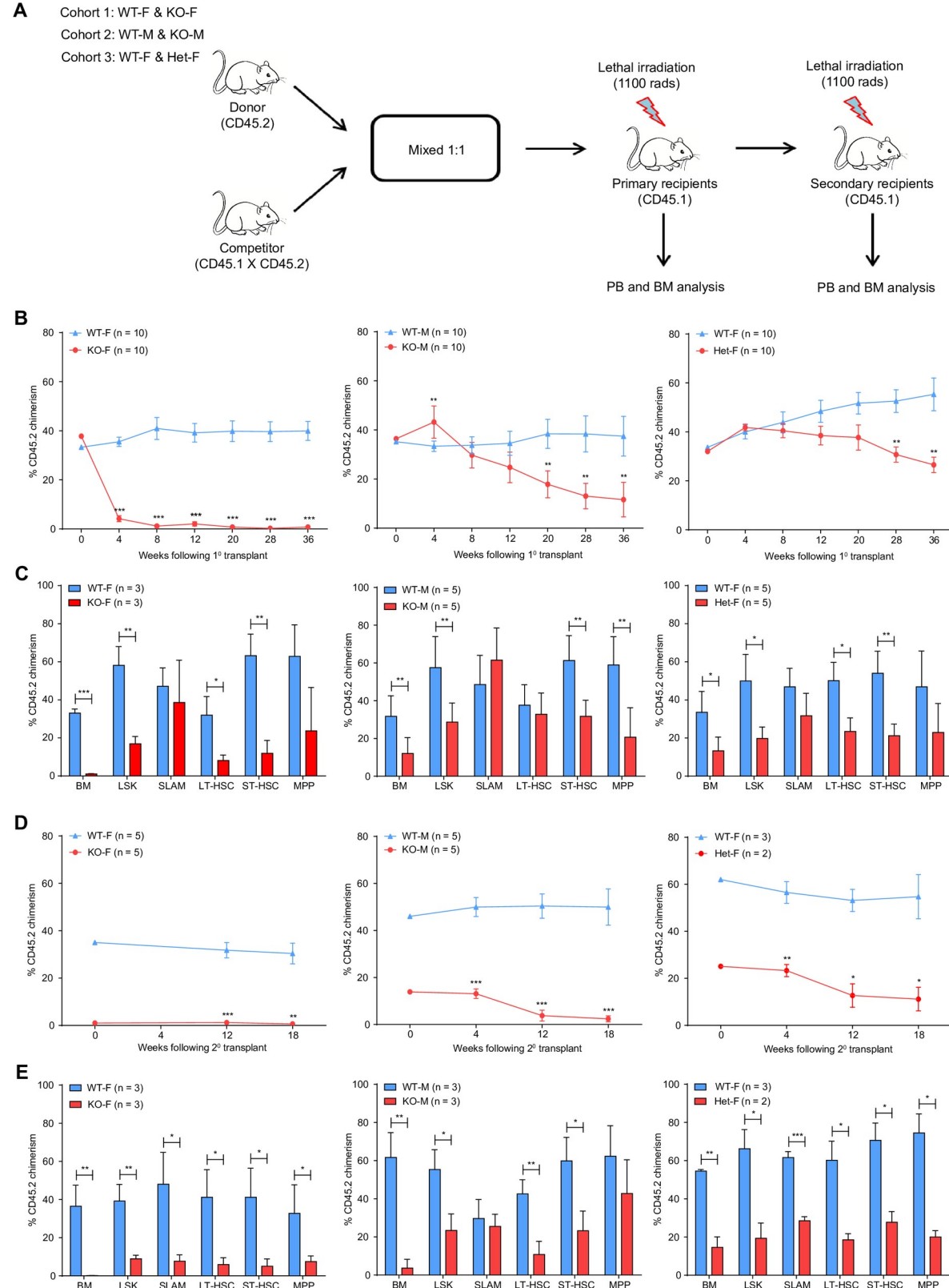

**Fig 4. *Kdm6a* conditional KO mice have decreased repopulating potential in serial competitive transplantation assays.** (A) Experimental scheme of a serial competitive repopulation assay. Whole BM cells were harvested from all cohorts of young *Kdm6a*

conditional KO mice (CD45.2) and mixed with WT competitor marrow (CD45.1 x CD45.2) in a 1:1 ratio, which was then transplanted into lethally-irradiated primary recipient mice (CD45.1). PB was examined for donor cell chimerism at indicated time points after transplantation. After 36 weeks of observation, BM cells from primary recipient mice of the same genotype were pooled equally and transplanted into lethally-irradiated secondary recipient mice. (B) PB donor cell chimerism of primary recipient mice. The respective curves showed a decreased repopulation potential for all three cohorts of *Kdm6a* conditional KO mice, which was most pronounced in KO-F mice. (C) Quantification of donor cell chimerism in hematopoietic stem cells (HSCs) in BM by flow cytometry 36 weeks after transplantation. A relative preservation of donor-derived cells in the SLAM compartment was found from all cohorts of *Kdm6a* conditional KO mice. The most striking difference was found in KO-F mice (35.51-fold increase in SLAM cells as compared to KO-F CD45.2$^+$ donor cells in whole BM, p = 0.0430). (D) PB donor cell chimerism of secondary recipient mice. (E) Quantification of donor cell chimerism of HSCs in BM by flow cytometry 18 weeks after secondary transplantation. Significant differences in the chimerism of SLAM cells compared to whole BM were found in KO-F mice (55.2-fold increase in CD45.2$^+$ KO-F SLAM cells as compared to CD45.2$^+$ cells in the whole BM, p = 0.0163) and KO-M mice (11.31-fold increase in CD45.2$^+$ KO-M SLAM cells as compared to CD45.2$^+$ cells in the whole BM, p = 0.0010). Error bars represented mean ± s.d. $^*$p<0.05, $^{**}$p<0.01, $^{***}$p<0.001 by unpaired two-tailed Student's *t*-test.

features [42–44]. We identified 14,892 enhancer-like regions with H3K4me peaks, and DNA accessibility determined by ATAC-seq, excluding promoter-proximal areas near annotated TSS or with high enrichment of H3K4me3. We categorized enhancer-like regions as "candidate-active" enhancer regions (5,559 regions) by including the concomitant presence of H3K27Ac peaks, or as "candidate-poised" enhancer regions (204 regions) by including the presence of H3K27me3 peaks. When comparing the candidate active enhancer regions of WT-F and KO-F mice, the samples were similar in both hierarchical heatmaps and peak density plots (**S7E Fig**).

We performed a statistical test on differential histone modifications. Samples did not group by genotype on a correlation matrix heatmap using the Pearson correlation coefficient. We found differential changes in H3K27me3, H3K27Ac, H3K4me3, H3K4me, and chromatin accessibility at 25, 440, 212, 110, and 76 promoters, respectively (FDR < 0.05) (**S2 Table**). At enhancers, we found that these chromatin modifications did not cluster by genotype, with the exception of H3K27Ac, which did group by genotype on a correlation matrix heatmap. We identified differential changes in H3K27me3, H3K4me, and chromatin accessibility at 19, 2, and 2 candidate poised enhancers respectively. Additionally, we found differential H3K27Ac and H3K4me peaks at 188 and 166 candidate active enhancers (**S3 Table**). Therefore, Kdm6a loss led to changes in chromatin modifications at a small number of specific loci, but did not widely disrupt the structure of chromatin throughout the genome.

### *Kdm6a* deficiency alters gene expression in hematopoietic cells

We used scRNA-seq on unfractionated BM cells of 4 young (<12-week-old) female mice (2 WT and 2 littermate matched *Kdm6a* null mice expressing Vav1-Cre) to infer hematopoietic cell lineages based on gene expression [45, 46], and to define differentially expressed genes (DEGs). BM from KO-F mice have an increased fraction of mature myelomonocytic cells, which include polymorphonuclear neutrophils (PMNs), monocytes, and macrophages. The largest contribution to the increase in myeloid cells appears to be from monocytes. In contrast, percentages of B- and T-cells were decreased in KO-F mice (**Fig 6A and 6B**).

To identify target genes regulated by *Kdm6a* in young KO-F hematopoietic cells, we first identified DEGs in pooled cells from all populations, and found 154 DEGs (of which 128 genes were downregulated) (**S4 Table**; **Fig 6C**). We characterized populations with the gene ontology (GO) annotation dataset to identify pathways enriched in the DEGs of whole BM. Downregulated genes in whole BM showed significant enrichment of immune pathways involved in antigen processing and T cell differentiation (**Fig 6D**). To filter for genes involved in myeloid differentiation, we compared whole BM DEGs with the myeloid cell differentiation GO gene set, and identified downregulation of *Cd74*, *Ccr7*, *Itgb3*, and upregulation of *Fam20c* (**Fig 6E**). As noted above, we detected very few changes in histone modifications in *Kdm6a* deficient

**A**

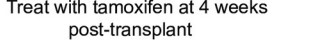
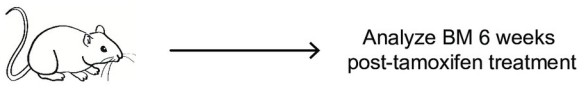

| Donor (CD45.2) | | | Recipient (CD45.2) | |
|---|---|---|---|---|
| Cohort | Genotype | | Cohort | Genotype |
| 1 | KO-F Cre-ERT2$^{+/-}$ | $1 \times 10^6$ cells $\rightarrow$ | 1 | KO-M Cre-ERT2$^{+/-}$ |
| 2 | KO-F & KO-M Cre-ERT2$^{+/-}$ | $\rightarrow$ | 2 | WT-F & WT-M Cre-ERT2$^{-/-}$ |
| 3 | WT-F & WT-M Cre-ERT2$^{-/-}$ | $\rightarrow$ | 3 | KO-M Cre-ERT2$^{+/-}$ |
| 4 | WT-F & WT-M Cre-ERT2$^{+/-}$ | $\rightarrow$ | 4 | WT-M Cre-ERT2$^{-/-}$ |

**B**

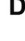

Treat with tamoxifen at 4 weeks post-transplant

$\rightarrow$ Analyze BM 6 weeks post-tamoxifen treatment

| Cohort | BM cells | Stromal cells |
|---|---|---|
| 1 | *Kdm6a*-null | *Kdm6a*-null |
| 2 | *Kdm6a*-null | *Kdm6a*-WT |
| 3 | *Kdm6a*-WT | *Kdm6a*-null |
| 4 | *Kdm6a*-WT | *Kdm6a*-WT |

**C**

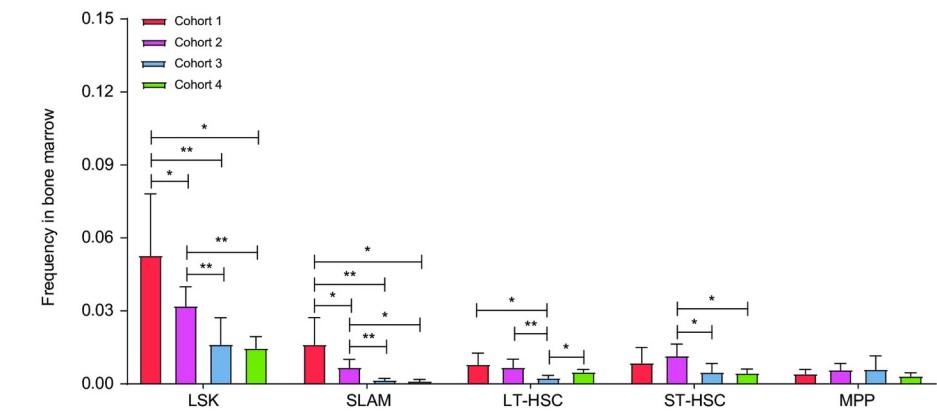

**D**

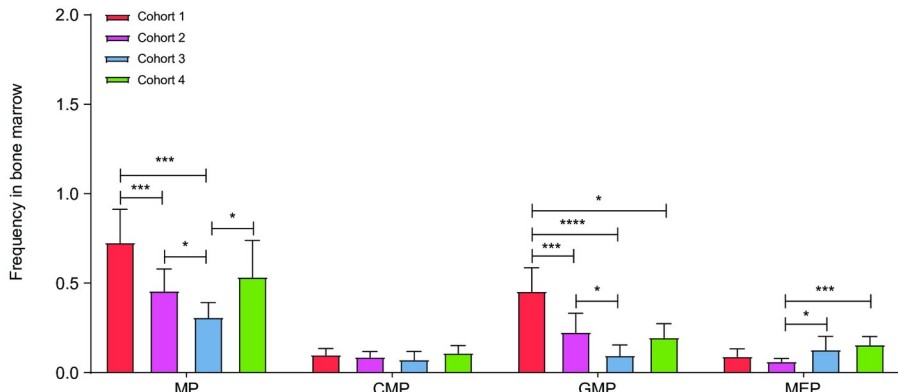

**Fig 5. The *Kdm6a*-null stroma promotes the expression of *Kdm6a*-null hematopoietic stem and progenitor cells.** (A and B) Experimental scheme of a non-competitive BM repopulation assay. (A) Whole BM cells from indicated donor mice (CD45.2) were transplanted into lethally-irradiated indicated recipient mice (CD45.2). (B) Recipient mice were given tamoxifen after 4 weeks of

engraftment to overcome engraftment defect. The Tamoxifen dosing protocol used was 3 mg/day/mouse, MWF x 3 weeks. 6 weeks after tamoxifen treatment, BM cells were examined by flow cytometry and the percentage of *Kdm6a* floxed alleles was measured by q-PCR. Table shows the expected genotype for the BM and stroma cells of the corresponding cohort. (C) Quantification of the indicated stem cell compartments showed a significant increase in LSK and SLAM cell populations in cohort 1 as compared to all other cohorts. (D) Quantification of the indicated progenitor compartments showed an increase in GMPs and a decrease in MEPs in cohort 1 as compared to all other cohorts. Error bars represented mean ± s.d. *p<0.05, **p<0.01, ***p<0.001, ****p<0.0001 by unpaired two-tailed Student's *t*-test.

BM cells. The most significant changes were in H3K27Ac peaks and H3K4me3 peaks near promoters, and H3K4me peaks and H3K27ac peaks near enhancers. When we compared the list of significantly modified genes from ChIP-seq and scRNA-seq datasets, we found only 9 overlapping genes in promoter regions, and 3 overlapping genes (*Cd2*, *Dapk2*, and *Edaradd*) at enhancer regions (**Fig 6F**).

Restricting the analysis to PMNs revealed 161 DEGs, of which 125 were downregulated. In monocytes, we identified 76 DEGs, of which 60 genes were downregulated (**S4 Table**). We noted a significant increase in the promyelocyte markers *Mpo* and *Ctsg* within the PMN population. The most significant DEG in whole BM cells was *Vcan*, a monocyte marker that was upregulated in monocyte-specific DEGs (**Fig 7A–7C**). PMN and monocyte populations were characterized with the GO annotation dataset. We noted that upregulated PMN genes were associated with an enrichment in neutrophil migration pathways; downregulated monocyte genes were associated with antigen processing and presentation pathways (**Fig 6D**). In PMNs, we identified downregulation of *Cd74*, *Flvcr1*, *Itgb3* and upregulation of *Ccl3*, *Fam20c*, and *Prtn3* in genes of the myeloid cell differentiation GO gene set. Using the same GO gene set, the monocyte population displayed reduced expression of *Cd74*, *Irf4*, *Itgb3*, and upregulation of *Hoxa7* (**Fig 7A and 7B**).

Using scRNA-seq data, we identified a decrease in the fraction of early stem/progenitor cells and an increase in GMPs of KO-F BM samples (**Fig 7D**). We identified 211 DEGs in early stem/progenitor cells, and 82 DEGs in GMPs. When filtering for genes involved in myeloid differentiation, we identified 5 DEGs in early stem/progenitor cells: 4 genes were downregulated (*Gpr171*, *Lmo2*, *Meis1*, *Thra*), and 1 was upregulated (*Prkdc*). In the GMP population, there were 2 DEGs that overlapped with genes of the myeloid cell differentiation GO gene set: *Ndfip1* was upregulated and *Hoxa7* was downregulated (**Fig 7F**).

## Discussion

In this study, we analyzed a *Kdm6a* conditional knockout mouse model to define the role of *Kdm6a* in normal hematopoiesis. Although five previous studies have evaluated *Kdm6a* deficiency in mouse hematopoietic cells [1, 33–36], this is the first to describe the use of *Vav1*-Cre to selectively inactivate *Kdm6a* in the HSPCs of mid-gestational embryos; surprisingly, the hematopoietic phenotype in this model is less severe than the others. The Vav1-cre system utilized here has been shown to have little to no off-target recombination, with the possible exception of endothelial cells. Mice in this study had an MDS-like phenotype that was associated with increased self-renewal, and was accentuated with aging. Further, *Kdm6a* inactivation induced by *Vav1*-Cre leads to a decreased repopulating potential that is most pronounced in the BM cells of homozygous female mice. Female *Kdm6a* KO mice developed an age-dependent expansion of HSPCs, which did not properly differentiate into mature cell types. Finally, we showed that concurrent *Kdm6a* inactivation in both BM stromal cells and hematopoietic cells exaggerated HSPC expansion, suggesting a role for *Kdm6a* in maintaining the BM microenvironment. These data may provide a partial explanation for why *Kdm6a* inactivation in

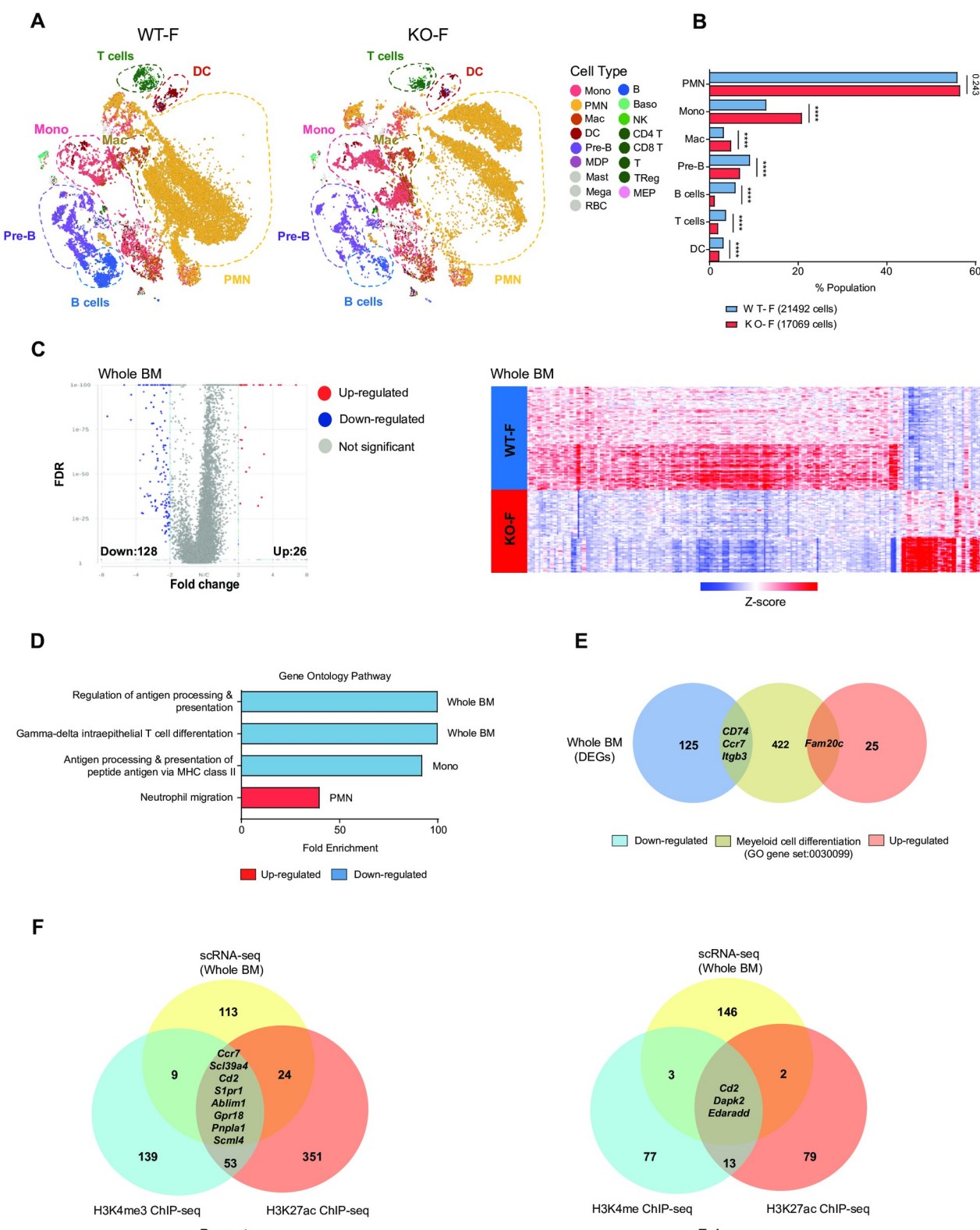

**Fig 6. The loss of *Kdm6a* alters gene expression in normal hematopoiesis.** (A) t-distributed stochastic neighbor embedding (t-SNE) plots of single cell RNA-seq (scRNA-seq) data from whole BM cells of WT-F mice (n = 2, total cells: 21492) and KO-F mice (n = 2, total cells: 17069). (B) Population fractions in whole BM cells from KO-F and WT-F mice associated with the scRNA-seq data as defined by Haemopedia gene expression. Cell populations shown include: PMNs, monocytes, macrophages, Pre-B cells, B cells, T cells and DCs. (C) Volcano plots of expression changes of

KO-F vs. WT-F from whole BM (FDR < 0.01, FC: > 2 and <-2). Heatmap of z-score values of 154 differentially expressed genes (DEGs) from KO-F mice compared to WT-F mice by scRNA-seq. (D) Gene ontology (GO) pathway analysis for all DEGs in whole BM, PMNs, and monocytes.(E) Venn diagram showing the overlap between all DEGs of whole BM and the Myeloid cell differentiation GO gene set (0030099). (F) Venn diagram showing the overlap between all DEGs of the whole BM scRNA-seq and Chip-seq data for H3K4me3 and H3K27ac promoters (left panel) or H3K4me and H3K27ac enhancers (right panel). Error bars represented mean ± s.d. ****p<0.0001 by Fisher's exact test with multiple hypothesis corrections.

both compartments may have a more severe phenotype in the previously reported models, with the frequent development of myeloid malignancies.

Five previous studies have reported the consequences of *Kdm6a* inactivation in hematopoietic cells [1, 33–36]. The strategies used to create these models, and a summary of their phenotypes, are shown in **Table 1**. In Thieme *et al*. [33], the *Kdm6a* floxed allele was the same as this study, but floxing was initiated with tamoxifen administration in mice bearing the Rosa-Cre-ERT2 allele, which is expressed ubiquitously in hematopoietic and stromal cells. Five weeks after tamoxifen administration, homozygous females exhibited splenomegaly, non-fatal anemia, thrombocytopenia, and leukopenia; hemizygous male mice had no detectable abnormalities. None of these mice were reported to develop leukemias, but they were not followed long-term. These authors also evaluated the marrow of mice two weeks after tamoxifen administration: they noted dysplasia of erythroid, megakaryocyte, and granulocyte lineages, and decreased CFU-GMs, BFU-Es, and CFU-Ms. In contrast, we noted an increase in CFU-GMs, and no evidence for leukopenia in the *Vav1*-Cre mouse model.

In the second study, Zheng *et al*. [1] inactivated *Kdm6a* by deleting exons 11–14, using Rosa-Cre-ERT2 mice and tamoxifen to induce somatic floxing; again, floxing occurs ubiquitously in these mice. In this non-transplanted mouse model, approximately 70% of mice (9/12 *Kdm6a* null female and 7/11 *Kdm6a* hemizygous males) developed a CMML-like disease with splenomegaly, monocytosis, and extramedullary hematopoiesis within 12 months after Tamoxifen administration. Hematopoietic-specific phenotypes were later established with transplantation, which demonstrated that *Kdm6a* loss leads to a deficiency in hematopoietic reconstitution, similar to findings reported here. Because Rosa26 is expressed ubiquitously, the inactivation of *Kdm6a* in all tissues (including both hematopoietic and stromal cells) may have made the myeloid phenotype in these mice more severe, as suggested in our evaluation of *Kdm6a* deficiency in out compartment-specific experiments.

Gozdecka *et al*. [34] demonstrated the novel finding that *Kdm6a* acts as a tumor suppressor, but in a noncatalytic fashion. In that study, the same exon 3 *Kdm6a*^flox allele was used, but floxing was initiated by pIpC injection in mice expressing *Mx1*-Cre. Nearly two thirds of their non-transplanted, pIpC treated *Kdm6a*-KO female mice developed a transplantable form of AML within 22 months. Similar to what is reported here, these authors demonstrated an increase in lineage negative cells and GMP cells in young female mice, and a decrease in MEPs. BM cells from both studies showed an expansion of myeloid cells with aberrant self-renewal.

Sera *et al*. [35] described a mouse model that conditionally deleted *Kdm6a* exons 11 and 12, using tamoxifen inducible *ERT2-Cre*^+ mice. These mice developed an MDS-like phenotype that was confirmed with morphology and flow cytometry. Although these mice did not develop spontaneous hematopoietic malignancies, leukemias did arise in mice with additional mutations created with retrovirus-insertional mutagenesis, using a Moloney murine leukemia virus fused to a 4070A retrovirus (MOL4070A), that has been shown to promote the development of myeloid leukemias [47]. In their experiment, the most common integration sites were in or near *Sox4*, *Mecom*, *Osvpl1a*, *Notch1*, *Ikaros*, and *Tax1bp1*. Retroviral mutagenesis was initiated in neonatal female *Kdm6a* KO mice, hemizygous male *Kdm6a* KO mice, and female and male wild type mice. During a 250-day observation period, all male *Kdm6a* hemizygous

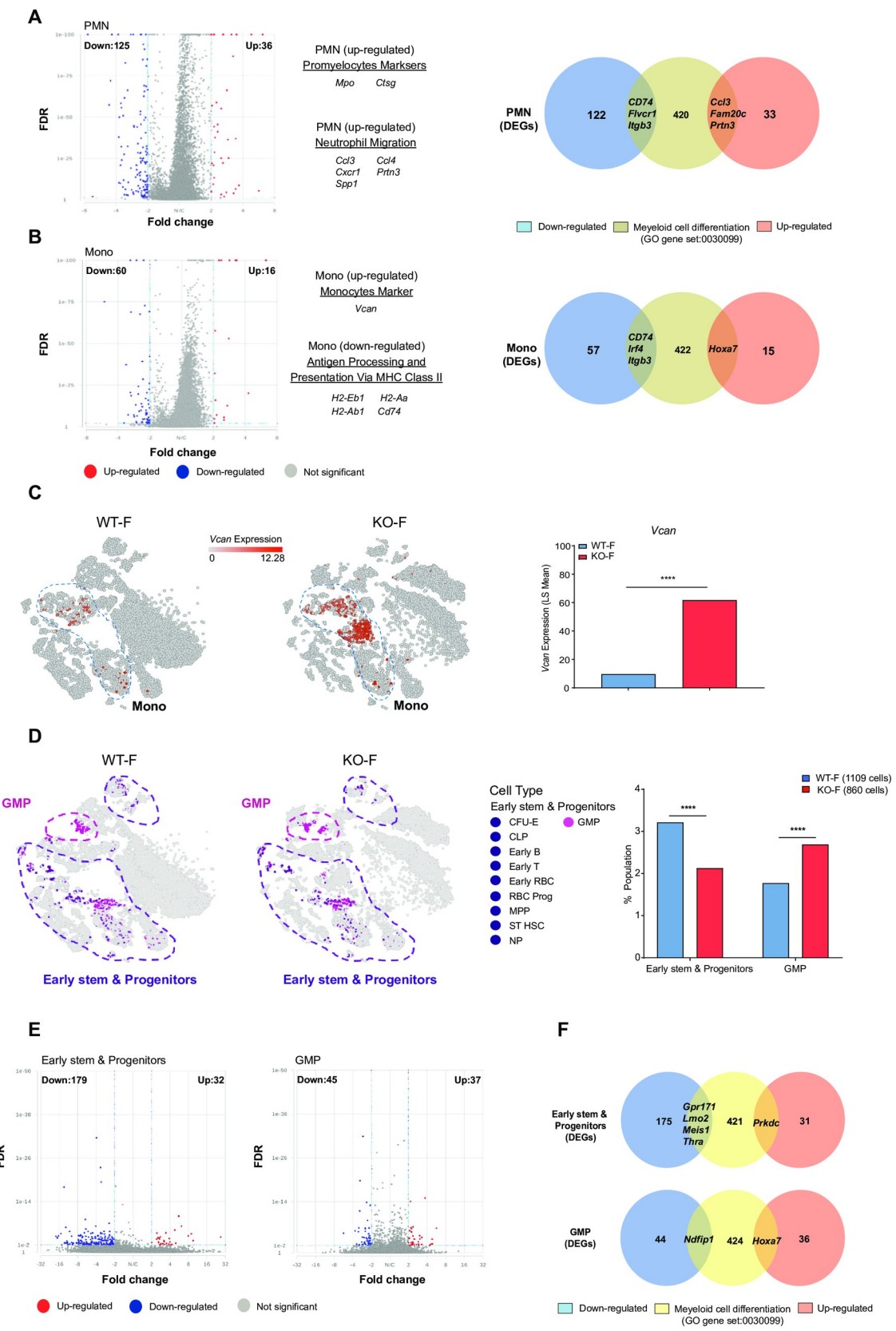

**Fig 7. The loss of *Kdm6a* alters gene expression in early stem/progenitor cells and mature myeloid cells.** (A) Volcano plot of expression changes from KO-F mice vs. WT-F mice in the PMN population at a FDR < 0.01, FC: > 2 and <-2 (left panel). Venn diagram showing the overlap between all 161 DEGs of the PMN population and the myeloid cell differentiation GO gene set (0030099) (right panel). (B) Volcano plot of expression changes from KO-F mice vs. WT-F mice in the monocyte population at a FDR < 0.01, FC: > 2 and <-2 (left panel). Venn diagram showing the overlap between all 76 DEGs of the monocyte population and the myeloid cell differentiation GO gene set (0030099) (right panel). (C) t-SNE plots showing relative expression of *Vcan* in whole BM cells from WT-F and KO-F mice and the relative expression value of *Vcan* from WT-F vs KO-F mice. (D) t-SNE plots showing stem and progenitor cells from WT-F mice (total cells:1109) and KO-F mice (total cells: 860). Early stem and progenitor cells in purple include CFU-E, early B cells, early RBCs, early T-cells, MPP, NP, RBC Prog, ST-HSC. GMP cells are shown in pink. Population fractions in whole BM cells from KO-F mice and WT-F mice associated with the scRNA-seq data. Data shows early stem and progenitor cells and GMP cells. (E) Volcano plots of expression changes for KO-F mice vs. WT-F mice from the early stem and progenitor population or GMP population. For both plots: FDR < 0.01, FC: > 2 and <-2. (F) Venn diagram showing the overlap between all DEGs of the early stem and progenitor population (211 total DEGs, top diagram) or GMP population (82 total DEGs, bottom diagram) and the myeloid cell differentiation GO gene set (0030099). Error bars represented mean ± s.d. ****p<0.0001 by Fisher's exact test with multiple hypothesis corrections.

MOL4070A mice died; all female *Kdm6a* KO MOL4070A mice died by day 150 (none of the WT mice died). Most of these mice developed AML (11/13 KO mice and 11/14 hemizygous mice), but cases of T-ALL (2/13 KO mice and 2/14 hemizygous mice) and B-ALL (1/14 hemizygous mice) were also described. Further, these authors described increased ROS, impaired DNA damage responses, and upregulation of multiple genes associated with aging in this model.

Finally, Huppertz *et al.* [36] recently described a role for *Kdm6a* in hematopoiesis and B-cell development under "stressed" conditions. They excised *Kdm6a* exon 3 utilizing *Mx1*-cre transgenic mice and pIpC administration, similar to that of Gozdecka *et al.* However, this study followed the mice for only 4–5 months post-pIpC treatment, and did not describe the development of any malignancies. These authors identified changes in the population sizes of LSKs, ST-HSCs, MPPs, CMPs, and GMPs that were different than those described in our model, which could potentially be due to the time at which these assessments were made (16–20 weeks, vs. <12 or >50 weeks in our study). Similar to our study, no significant changes were noted in the hematopoietic populations of hemizygous male mice. However, these authors noted an increase in pro-B cells and decrease of pre-B cells in *Kdm6a*-KO mice compared to WT. In a competitive transplant environment, the change in B-cell compartments was only noted after replicative stress was induced by 5-FU.

We have summarized the models and the key findings of these six studies (Table **1**) to highlight how important the experimental design can be for the outcomes of somatic gene inactivation studies. The results from these six studies have similarities and differences in phenotypes, with striking differences in the incidence of hematopoietic malignancies in *Kdm6a* deficient mice. In our study, none of the mice developed AML, even after long latent periods, while in other studies, many mice developed AML or a related malignancy. Although it is not clear what is causing these differences, factors that may contribute include subtle differences in mouse strains, the hematopoietic and stromal compartments in which floxing occurs, the use of pIpC to induce floxing in the *Mx1*-Cre mice, the use of tamoxifen to induce floxing in the ERT2-Cre mice, and differences in the environments in which mice were housed, among others. The importance of these factors is underscored by the fact that four of these studies (including ours) used ES cells with an identical excision of *Kdm6a* exon 3. [39] Clearly, one or more of these factors is highly relevant for the progression to AML (and other phenotypes) in this model system; additional studies will be required to define the nature and importance of these factors.

To study the role of the catalytic function of Kdm6a in hematopoietic cells, we used CHIP-seq to examine global changes in histone methylation and acetylation, using an integrative approach to define all locations of histone modifications; no large-scale changes in histone

**Table 1. Summary of previous studies of *Kdm6a* inactivation in hematopoiesis.**

| Study | *Kdm6a* mutation | Cre-Strategy | Strain | Age at assessment | Progenitor Phenotype in BM | Malignancy | H3K27me3 levels | Phenotype: Female | Phenotype: male |
|---|---|---|---|---|---|---|---|---|---|
| **Thieme *et al.* 2013** | Deletion of exon 3 | Rosa-Cre-ERT2 + Tam | C57BL/6J | 5 weeks after Tam | ↓CFU-GMs ↓BFU-Es ↓CFU-Ms | None | Not assessed | Splenomegaly; Anemia; Thrombocytopenia; Leukopenia | None identified |
| **Zheng *et al.* 2018** | Deletion of exons 11–14 | Rosa-Cre-ERT2 +Tam | C57BL/6J | 10 months after Tam | ↑ LT-HSCs; ↑ ST-HSCs; ↑ GMPs; ↓ MEPs | CMML development within 12 months | Not assessed | Splenomegaly; Monocytosis; Extramedullary hematopoiesis | Development of CMML at a lower penetrance |
| **Gozdecka *et al.* 2018** | Deletion of exon 3 | *Mx1*-Cre + pIpC | C57BL/6 | 4–5 weeks after pIpC | ↑ LT-HSCs; ↑ ST-HSCs; ↑ GMPs; ↑ CMPs; ↓ MEPs; ↓ CLPs | AML within 22 months in females only | No large-scale changes identified | Thrombocytopenia | ↓ MEPs; ↓ CLPs; No leukemia developed |
| **Sera *et al.* 2021** | Deletions of exons 11–12 | Rosa-Cre-ERT2 + Tam | C57BL/6N | Not specified | ↑ LT-HSCs; ↑ ST-HSCs | Yes, with retroviral mutagenesis | Assessed in LT-HSCs and LSKs, in both increased | Thrombocytopenia; Increased WBC; Extramedullary hematopoiesis | ↑ WBC; ↑ CMPs |
| **Huppertz *et al* 2021** | Deletion of exon 3 | *Mx1*-Cre + pIpC | C57BL/6A | 4–5 months after pIpC | ↑ ST-HSCs; ↑ CLPs; ↑ CMPs; ↓ MEPs | None | No changes identified | Not reported | No changes identified |
| **Tian *et al* (this study)** | Deletion of exon 3 | Vav1-Cre | C57BL/6J | young (<12 weeks), aged (50- to 55-weeks) | Young: ↓LSK; ↓ MPP; ↑ GMPs; ↓ MEPs; Aged: ↑ SLAM; ↓ MPP; ↓ MEPs | None | No large-scale changes identified | Young: Anemia; Thrombocytopenia; Splenomegaly; Extramedullary hematopoiesis; Aged: Neutrophilia; Lymphopenia; Thrombocytopenia; Extramedullary hematopoiesis | No changes seen at the young or aged time periods |

methylation and/or acetylation were detected. We did identify locus-specific changes, a finding similar to that of Gozdecka *et al*, who performed Chip-seq on unfractionated BM samples, and also on sorted HSPCs. Like Gozdecka *et al.* and Huppertz *et al.*, we did not find large scale changes in H3K27me3, supporting a non-catalytic, alternative role for *Kdm6a* in leukemia development.

Using scRNA-seq data, we observed that 83% of DEGs in whole BM are downregulated, supporting a role for *Kdm6a* as a transcriptional activator. Although previous studies have suggested that *Kdm6a* can be a transcriptional activator or repressor, these results were based on

bulk RNA-seq of HSPCs [34]. scRNA-seq allowed us to define population-specific changes, and we identified many downregulated genes in stem and progenitor cells, PMNs, and monocytes, further suggesting that *Kdm6a* may normally act as a transcriptional activator in these compartments. Gozdecka *et al*. described RNA-seq of sorted HSCs, excluding DEGs found in male mice at an FDR value of less than 0.05 and a FC: > 0.5 and <-0.5. Using these cutoffs, they identified 2,886 DEGs, of which approximately half were downregulated (compared to 179 downregulated genes in early stem/progenitor cells in our data); 36 of these downregulated genes were found in both datasets, and only one common gene was upregulated (*Slc41a1)*. Additional studies will be required to better understand the roles of DEGs and dysregulated pathways in Kdm6a deficient HSPCs and myelomonocytic cells.

Our studies have demonstrated that the inactivation of *Kdm6a* in mice expressing *Vav1*-Cre leads to the development of an MDS-like disease that is age and sex dependent. Our data suggest that *Kdm6a* may act as a context-dependent tumor suppressor that is important for maintenance of normal myeloid development, that when disrupted, may contribute to the pathogenesis of AML.

## Supporting information

**S1 Fig. Characterization of the Het-F and KO-M mice.** (A) CBCs for the young and aged KO-M mice versus WT-M mice showed no significant differences, except in aged mice which had an increase in MCV. (B) CBCs for the young and aged Het-F mice versus WT-F mice showed no significant differences. (C) Splenomegaly was present in young KO-F mice (0.13 g in KO-F mice versus 0.06 g in WT-F mice, p = 0.0001) and aged KO-F mice (0.212 g in KO-F mice versus 0.085 g in WT-F mice, p = 0.0062). We also observed mild splenomegaly in young KO-M mice (0.11 g in KO-M mice versus 0.07 g in WT-M mice, p = 0.0102), but no significant differences between other matched cohorts. (D) Histopathologic examination of BM and spleen by H&E staining. BM sections showed moderately to markedly increased numbers of myeloid cells and decreased numbers of erythroid cells from young to aged KO-F mice. The myeloid to erythroid ratio was typically elevated, up to 20:1 in some fields. Emperipolesis was often noted. Megakaryocytes were present in expected to increased numbers. Sections of spleen from KO-F mice showed extramedullary erythropoiesis and granulopoiesis that increased with age. Representative images (40x magnification) including small fields (100x magnification) are shown. Error bars represented mean ± s.d. *p<0.05, **p<0.01, ***p<0.001 by unpaired two-tailed Student's *t*-test.
(PDF)

**S2 Fig. Flow cytometry characterization of Het-F and KO-M mice.** (A) Cell lineage frequencies for BM, spleen and PB from young and aged, KO-M and WT-M mice were determined by flow cytometry for myeloid cells, B-cells, T-cells, NK cells and erythroid precursors. There was a significant decrease in B220$^+$ B cells in the spleen of young KO-M mice as compared to WT-M mice. (B) Cell lineage frequencies for BM, spleen and PB from young and aged, Het-F and WT-F mice were determined by flow cytometry for myeloid cells, B cells, T cells, NK cells and erythroid precursors. There were no significant differences observed in these compartments. Error bars represented mean ± s.d. *p<0.05 by unpaired two-tailed Student's *t*-test.
(PDF)

**S3 Fig. The inactivation of *Kdm6a* in mice had age- and gender-dependent effects on primitive hematopoietic cell populations and myeloid progenitors [48].** Flow cytometry analysis of HSPC populations in young and aged KO-M and Het-F mice (compared with appropriate littermate controls). Quantification of the indicated stem cell and progenitor compartments

showed no differences for these genotypes at any time point.
(PDF)

**S4 Fig. *Kdm6a* conditional KO mice had a decreased repopulation potential in serial competitive transplantation.** (A) Quantification of donor cell chimerism in mature cell lineage and myeloid progenitor cell populations each in the PB and BM analyzed by flow cytometry 36 weeks after transplantation showed a nearly complete loss of KO-F donor cells in these cell populations. (B) Quantification of donor cell chimerism in mature cell lineage and myeloid progenitor cell populations each in the PB and BM analyzed by flow cytometry 36 weeks after transplantation showed a significant decrease of KO-M donor cells in these populations. (C) Quantification of donor cell chimerism in mature cell lineage and myeloid progenitor cell populations each in the PB and BM analyzed by flow cytometry 36 weeks after transplantation showed a significant decrease of Het-F donor cells in these cell populations. Error bars represented mean ± s.d. $^*$p<0.05, $^{**}$p<0.01, $^{***}$p<0.001 by unpaired two-tailed Student's *t*-test.
(PDF)

**S5 Fig. The *Kdm6a*-null stroma and hematopoietic compartment did not show sex specific differences.** (A) Quantification of the indicated stem cell compartments for cohorts 1–4, separated by sex. (B) Quantification of the indicated progenitor compartments for cohorts 1–4, separated by sex.
(PDF)

**S6 Fig. Characterization of 18-month-old *Kdm6a* conditional KO x *Vav1*-Cre mice.** (A) The normal survival curve demonstrated that the inactivation of *Kdm6a* could not induce mice to develop overt leukemia or die of lethal abnormalities. (B) Spleen weight for all genotypes of *Kdm6a* conditional KO mice and appropriate littermate controls. Several KO-F mice had marked splenomegaly (> 0.5 g). (C) CBCs showed that KO-F mice had anemia and thrombocytopenia, but no difference was seen in Het-F or KO-M mice. (D) Flow cytometry analysis of mature cell lineage in the spleen from KO-F and WT-F mice. Representative flow cytometry plots showed the deletion of *Kdm6a* caused a striking myeloid skewing with an increased number of Gr1$^+$CD11b$^+$ myeloid cells and an increase in Ter119$^+$CD71$^+$ erythroid precursors with age. (E) Experimental scheme of the KO-F spleen cells transplantation. Spleen cells were harvested from 3 18 months old KO-F donor mice and transplanted into non-irradiated recipient mice. PB was examined for donor cell chimerism on a monthly basis after transplantation. (F) PB donor cell chimerism indicated the donor-derived spleen cells could not engraft in recipient mice. After long-term observation, mice from each donor cohort were sacrificed and BM cells were harvested for flow cytometry analysis. Representative flow cytometric plots and quantification of donor cell chimerism in whole BM and the HSC compartment (LSK). KO-F donor HSC cells engraft and exhibited a great contribution to the SLAM compartment (14.6% in LSK cells and 50% in SLAM cells). Error bars represented mean ± s.d. $^*$p<0.05, $^{**}$p<0.01, $^{***}$p<0.001 by unpaired two-tailed Student's *t*-test.
(PDF)

**S7 Fig. Global identification and quantification of H3K4me3, H3K4me, H3K27me3 and H3K27ac binding profiles at promoter and enhancer-like regions in WT-F and KO-F mouse models.** (A) The detection and quantification of 21 different histone H3 modifications using total core histone proteins extracted from WT-F and KO-F mice whole BM cells. (B) The detection and quantification of 10 different histone H4 modifications using total core histone proteins extracted from WT-F and KO-F mice whole BM cells. (C) Western blot analysis of core histone protein lysates extracted from whole BM cells showed a decrease of H3K27 acetylation in the young KO-F mice compared with WT-F mice. (D) Peak density heat maps

for each ChIP-seq dataset, ATAC-seq and bulk RNA-seq around the transcription start sites (TSSs) of the complete set of 18,624 potentially transcriptionally-engaged promoters of annotated genes defined by our own H3K4me3 ChIP-seq dataset. Total peaks detected for H3K27ac were 33,464 peaks (139,420,356 reads) for WT-F samples and 28,290 peaks (146,408,232 total reads) for KO-F samples. Tag densities within a window of +/− 10 kb on either side of TSS coordinates were collected from each dataset. We observed the expected inverse relationship between H3K4me3 and H3K27me3 peak density that correlates with transcriptional activity of annotated genes (defined by bulk RNA-seq data) (top panel). Peak density profiles around TSS (+/− 4kb) for each ChIP-seq dataset with ATAC-seq and bulk RNA-seq (lower panel). (E) Peak density heat maps for each ChIP-seq dataset and ATAC-seq around enhancer-like regions of the complete set of 14,892 enhancer-like regions by presence of H3K4me peaks and ATAC-seq. Tag densities within a window of +/− 10 kb on either side of the enhancer coordinates were collected from each dataset (top panel). The peak density profiles around enhancers (+/− 4kb) for each ChIP-seq dataset with ATAC-seq (lower panel). Error bars represented mean ± s.d. $^*$p$<$0.05 by unpaired two-tailed Student's $t$-test.
(PDF)

**S8 Fig. Distribution of genome-wide ChIP-seq templates around the annotated TSS.** (A) H3K27me3 ChIP-seq (n = 3 per genotype). (B) H3K27ac ChIP-seq (n = 2 per genotype). (C) H3K4me ChIP-seq (n = 2 per genotype). (D) H3K4me3 ChIP-seq (n = 3 per genotype). (E) ATAC-seq (n = 2 per genotype). ChIP-seq and ATAC-seq using bulk BM cells derived from young, KO-F mice versus young, WT-F mice as controls.
(PDF)

**S1 Table. List of antibodies used for flow cytometry with appropriate catalog numbers.**
(XLSX)

**S2 Table. Promoter list of differential H3K4me3 modification between WT and KO.**
(XLSX)

**S3 Table. Table of active enhancers of differential modification between Kdm6a-WT and KO at H3K4 and H3K27.**
(XLSX)

**S4 Table. Table of differentially expressed genes between WT and KO for bulk bone marrow, progenitors, GMPs, PMNs, monocytes, and macrophages.**
(XLSX)

**S1 Methods. Additional information regarding experimental procedures and material can be found in S1 Methods file.**
(DOCX)

**S1 Raw images.**
(PDF)

## Acknowledgments

The authors would like to thank Dr. Timothy J. Ley and Dr. Matthew Walter for their continued mentorship and help throughout the writing and review process. The authors would also like to thank Julie K. Ritchey for technical assistance with mouse BM transplants and Conner York for maintenance of our mouse colony.

## Author Contributions

**Conceptualization:** Ling Tian, Lukas D. Wartman.

**Data curation:** Gue Su Chang, Christopher A. Miller.

**Formal analysis:** Ling Tian, Monique Chavez, Gue Su Chang, Lukas D. Wartman.

**Funding acquisition:** Christopher A. Miller, Lukas D. Wartman.

**Investigation:** Ling Tian, Monique Chavez, Nichole M. Helton.

**Methodology:** Ling Tian, Lukas D. Wartman.

**Project administration:** Lukas D. Wartman.

**Resources:** Lukas D. Wartman.

**Software:** Gue Su Chang, Christopher A. Miller.

**Supervision:** Lukas D. Wartman.

**Validation:** Ling Tian, Monique Chavez.

**Visualization:** Ling Tian, Monique Chavez, Gue Su Chang, Lukas D. Wartman.

**Writing – original draft:** Ling Tian, Monique Chavez, Casey D. S. Katerndahl, Lukas D. Wartman.

**Writing – review & editing:** Ling Tian, Monique Chavez, Casey D. S. Katerndahl, Lukas D. Wartman.

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
