## [Decision Letter · Decision Letter 0]

27 Aug 2021

PONE-D-21-23013

Kdm6a Deficiency Restricted to Mouse Hematopoietic Cells Causes an Age- and Sex-dependent Myelodysplastic Syndrome-Like Phenotype

PLOS ONE

Dear Dr. Wartman,

Thank you for submitting your manuscript to PLOS ONE. Your study has now been evaluated by two reviewers. As you will see both reviewers find your study of interest but also raise several points that would need to be addressed. Therefore, I invite you to submit a revised version of the manuscripts. If you were to provide me with a point to point response on how you have addressed all concerns raised by the reviewers inyour revised manuscript, I would be in a strong position to make a decision on publication or your study in PLoS ONE.

We look forward to receiving your revised manuscript.

Kind regards,

Anton Wutz

Academic Editor

PLOS ONE

Journal Requirements:

Reviewers' comments:

Reviewer's Responses to Questions

**Comments to the Author**

1. Is the manuscript technically sound, and do the data support the conclusions?

Reviewer #1: Yes

Reviewer #2: Yes

2. Has the statistical analysis been performed appropriately and rigorously? 

Reviewer #1: Yes

Reviewer #2: Yes

3. Have the authors made all data underlying the findings in their manuscript fully available?

Reviewer #1: Yes

Reviewer #2: Yes

4. Is the manuscript presented in an intelligible fashion and written in standard English?

Reviewer #1: Yes

Reviewer #2: Yes

5. Review Comments to the Author

Reviewer #1: The histone H3K27me2/3-specific demethylase Kdm6a is mutated in many different human malignancies among them various blood cancers. Over the past years, several research groups have analysed the consequences of conditional Kdm6a deletion for adult haematopoiesis and tumorigenesis in the mouse. Tian and colleagues used the vav1-cre deleter strain, which has so far not been used for conditional mutation of Kdm6a in the hematopoietic system. The results reported in this study partially overlap with previously published findings but also add novel and potentially important new data, which make a significant addition to the knowledge published so far. The experiments are solid and convincing and the data support the conclusions. Nevertheless, I have a few comments that should be addressed before publication:

1) The vav1-cre transgene is not only active in hematopoietic cell types but also in endothelial cells, which potentially contribute to the hematopoietic niche. Therefore, the statement that only hematopoietic cells are targeted in the reported experimental setup should be toned down.

2) Results section page 11 line 265: The authors state that in the competitive transplantation assays Het-F and KO-M donor cells had a significant competitive disadvantage compared to WT competitor cells but did not have cell migration or engraftment problems. As no specific migration or homing assay was performed to support this statement, the authors should explain in a bit more detail how they came to this conclusion.

3) One of the most important findings described in the current manuscript is that Kdm6a mutant bone marrow stroma promotes the survival of hematopoietic progenitor cells. Unfortunately, the set of experiments that led to this conclusion is not explained in sufficient detail: i) given the differences in the male and the female HSPC compartment, it is rather surprizing that in the non-competitive transplantation experiments (Fig.5) male and female donors were combined in one cohort (cohorts 2-4). What was the rationale behind the combined analyses in this set of experiments? ii) Why did the authors choose to combine male and female recipients in cohort 2 while in the other cohorts only male recipients were used?

Typos:

Abstract: In line 26 it should read histone H3 K27me3.

Discussion: In line 599 it should read …Kdm6a may normally act as a transcriptional activator…

Reviewer #2: This paper presents an important refinement of previous studies to look at the role of KDM6a in haemopoeisis and myelodysplastic syndrome. Interestingly, they find a female sex bias in susceptibility to MDS and dysregulation of haempoiesis. Their analysis is extensive but does not reveal the epigenetic basis for the phenomenon they describe. Nevertheless, the work is an important contribution and seems well executed. Although KDM6a is an X-linked gene it is not clear from the present study what the underlying molecular mechanisms are, however, it is important that this work is published as it contributes to a further understanding of the generation of AML and other haemopoietic disorders. I would suggest toning down the last sentence of the abstract: "and that its inactivation may

contribute to AML pathogenesis by altering the epigenetic state of HSPCs ." as no clear-cut change in epigenetic state has been identified as far as I can see.

6. PLOS authors have the option to publish the peer review history of their article (what does this mean?). If published, this will include your full peer review and any attached files.

Reviewer #1: No

Reviewer #2: No

---

## [Author Response · Author response to Decision Letter 0]

4 Oct 2021

1) We note that you have included the phrase “data not shown” in your manuscript. Unfortunately, this does not meet our data sharing requirements. PLOS does not permit references to inaccessible data. We require that authors provide all relevant data within the paper, Supporting Information files, or in an acceptable, public repository. Please add a citation to support this phrase or upload the data that corresponds with these findings to a stable repository (such as Figshare or Dryad) and provide and URLs, DOIs, or accession numbers that may be used to access these data. Or, if the data are not a core part of the research being presented in your study, we ask that you remove the phrase that refers to these data.

a) We thank the Editor for bringing this to our attention. The necessary data has been provided in the text, so the phrase “data not shown” is not necessary and we have removed it in all three instances. 

Reviewer 1

The histone H3K27me2/3-specific demethylase Kdm6a is mutated in many different human malignancies among them various blood cancers. Over the past years, several research groups have analysed the consequences of conditional Kdm6a deletion for adult haematopoiesis and tumorigenesis in the mouse. Tian and colleagues used the vav1-cre deleter strain, which has so far not been used for conditional mutation of Kdm6a in the hematopoietic system. The results reported in this study partially overlap with previously published findings but also add novel and potentially important new data, which make a significant addition to the knowledge published so far. The experiments are solid and convincing and the data support the conclusions. Nevertheless, I have a few comments that should be addressed before publication:

1) The vav1-cre transgene is not only active in hematopoietic cell types but also in endothelial cells, which potentially contribute to the hematopoietic niche. Therefore, the statement that only hematopoietic cells are targeted in the reported experimental setup should be toned down.

a) We thank the reviewer for raising this important point. While there is evidence that endothelial cells show recombination in the Vav1-Cre mouse strain, there is no evidence of Vav1-regulated transgenes in adult mice so the exact mechanism for Vav1-Cre mediated recombination in endothelial cells is unknown [1]. Unlike Georgiades et al. which identified recombination in endothelial cells, an independent study by Boer et al. did not see this pattern of expression and emphasized the hematopoietic specificity of the Vav1-Cre system [1,2]. However, we agree that it is important to acknowledge the findings presented by Georgiades et al., and to address the reviewers concern we have made sure to not over-emphasize the idea that only hematopoietic cells are targeted. The section now reads, “Since Vav1-Cre inactivates Kdm6aflox/flox alleles in HSPCs (but not other BM stroma cells, with the possible exception of endothelial cells), and since other models of Kdm6a knockouts inactivated the gene in both compartments, we wished to define the effect of stromal Kdm6a inactivation for hematopoietic phenotypes.” Finally, in the discussion the following sentence has been added, “The Vav1-Cre system utilized here has been shown to have little to no off-target recombination, with the possible exception of endothelial cells.”

2) Results section page 11 line 265: The authors state that in the competitive transplantation assays Het-F and KO-M donor cells had a significant competitive disadvantage compared to WT competitor cells but did not have cell migration or engraftment problems. As no specific migration or homing assay was performed to support this statement, the authors should explain in a bit more detail how they came to this conclusion.

a) We apologize for the lack of clarity in this section of text. We came to this conclusion because although we do not perform specific homing assays for the donor cells, homing is a necessary step for engraftment and reconstitution of the bone marrow [3] so the ability of donor cells (WT, KO-M, and Het-F) to reconstitute the bone marrow is showing homing into the proper bone marrow niches. We have also added a statement to address possible timepoints where engraftment defects are not measured: “However, there may have been cell migration and subtle engraftment problems at an early timepoint after transplantation that our current assay and measurements did not account for.”

3) One of the most important findings described in the current manuscript is that Kdm6a mutant bone marrow stroma promotes the survival of hematopoietic progenitor cells. Unfortunately, the set of experiments that led to this conclusion is not explained in sufficient detail: 

a) Given the differences in the male and the female HSPC compartment, it is rather surprizing that in the non-competitive transplantation experiments (Fig.5) male and female donors were combined in one cohort (cohorts 2-4). What was the rationale behind the combined analyses in this set of experiments?

i) Thank you for raising this question. For cohort 2-4 we combined the analyses because we did not see differences between male and female donors (of the same genotype) into male recipients. We felt that because there was a lack of sex specific changes it made the figure more complicated if we also separated each cohort by sex rather than just the bone marrow and stroma genotype. We have however now included a supplemental that separates the female and male donors. 

b) Why did the authors choose to combine male and female recipients in cohort 2 while in the other cohorts only male recipients were used?

i) We thank the reviewer for raising this important question. Similar to our previous results we chose to combine these data because we did not appreciate any biological differences in between the male and female recipients. Due to this lack of sex-specific differences we chose to combine the data as this provided a larger group of mice for the study. The added supplemental figure also separates the each cohort by recipient sex. 

4) Typos

Abstract: In line 26 it should read histone H3 K27me3.

Discussion: In line 599 it should read …Kdm6a may normally act as a transcriptional activator…

a) We apologize for the oversight and thank the reviewer for these corrections, we have addressed them in the manuscript.

Reviewer 2

This paper presents an important refinement of previous studies to look at the role of KDM6a in haemopoeisis and myelodysplastic syndrome. Interestingly, they find a female sex bias in susceptibility to MDS and dysregulation of haempoiesis. Their analysis is extensive but does not reveal the epigenetic basis for the phenomenon they describe. Nevertheless, the work is an important contribution and seems well executed. Although KDM6a is an X-linked gene it is not clear from the present study what the underlying molecular mechanisms are, however, it is important that this work is published as it contributes to a further understanding of the generation of AML and other haemopoietic disorders. 

1) I would suggest toning down the last sentence of the abstract: "and that its inactivation may contribute to AML pathogenesis by altering the epigenetic state of HSPCs." as no clear-cut change in epigenetic state has been identified as far as I can see.

a) We thank the reviewer for raising this critical point. We agree that based on our findings it is best that we tone down the last sentence of the abstract as we did not see large scale changes in H3K27me3. We have removed the reference to the “epigenetic state of HSPCs” from our manuscript abstract. 

References

1. Georgiades, P., Ogilvy, S., Duval, H., Licence, D. R., Charnock‐Jones, D. S., Smith, S. K., & Print, C. G. (2002). VavCre transgenic mice: a tool for mutagenesis in hematopoietic and endothelial lineages. genesis, 34(4), 251-256.

2. de Boer, J., Williams, A., Skavdis, G., Harker, N., Coles, M., Tolaini, M., ... & Kioussis, D. (2003). Transgenic mice with hematopoietic and lymphoid specific expression of Cre. European journal of immunology, 33(2), 314-325.

3. Srour, E. F., Jetmore, A., Wolber, F. M., Plett, P. A., Abonour, R., Yoder, M. C., & Orschell-Traycoff, C. M. (2001). Homing, cell cycle kinetics and fate of transplanted hematopoietic stem cells. Leukemia, 15(11), 1681-1684.

---

## [Editor Report · Decision Letter 1]

8 Oct 2021

PONE-D-21-23013R1Kdm6a Deficiency Restricted to Mouse Hematopoietic Cells Causes an Age- and Sex-dependent Myelodysplastic Syndrome-Like PhenotypePLOS ONE

Dear Dr. Wartman,

Thank you for submitting your revised manuscript to PLOS ONE. I have now read through the revised version and the response to the reviewers' comments. I find that you have addressed all earlier concerns. However, there are a few corrections needed before the study can be accepted ( Line 374 the figure S7A-C figure S6A). Therefore, we invite you to submit a further revised version of the manuscript that addresses the points raised during the review process. Please use this revision to check once more that the figures are referenced correctly in the text.

We look forward to receiving your revised manuscript.

Kind regards,

Anton Wutz

Academic Editor

PLOS ONE

Journal Requirements:

Additional Editor Comments (if provided):

The revised version has addressed the earlier concerns of the reviewers in a comprehensive manner. There are a few minor corrections that would be needed before the study can be accepted.

Minor points

a) Line 374 the figure S7A-C reference does not show an immunoblot of H3K27ac. Please, check the figure numbering and references in the text match up.

b) Figure S6A shows survival curves for different genotypes. The curves of all but the control (black) are not obvious which might be caused by overlap. Please, check that this figure is a correct representation of the data.
---

## [Author Response · Author response to Decision Letter 1]

13 Oct 2021

Dear Dr. Wutz, 

We want to take the time to thank you for your investment in the review process. Based on your careful assessment we have made the following changes: 

1) Line 374 the figure S7A-C reference does not show an immunoblot of H3K27ac. Please, check the figure numbering and references in the text match up.

We apologize for the oversight and thank the editor for bringing this inconsistency to our attention. After adding figure S5 to address reviewer comments our numbering did not adjust properly. In the current version all supplemental figure legends, text, and figure match up. 

B) Figure S6A shows survival curves for different genotypes. The curves of all but the control (black) are not obvious which might be caused by overlap. Please, check that this figure is a correct representation of the data.

We thank the editor for raising this point. For figure S6A there was no differences between in the leukemia free percent survival curves for each genotype. When these data are plotted on Prism Graph Pad, the overlap between genotypes causes there to be only one line observed. We have gone back to the original files and can confirm that these data include all genotypes. To make these data more clear we have changed the title of the Y-axis to “leukemia free percent survival” and have included censored data of mice that died of other causes not related to a leukemia or known malignancy.

---

## [Editor Report · Decision Letter 2]

20 Oct 2021

Kdm6a Deficiency Restricted to Mouse Hematopoietic Cells Causes an Age- and Sex-dependent Myelodysplastic Syndrome-Like Phenotype

PONE-D-21-23013R2

Dear Dr. Wartman,

thank you for sending your further revised manuscript. I am pleased to inform you that your manuscript has been judged scientifically suitable for publication and will be formally accepted for publication once it meets all outstanding technical requirements.

Kind regards,

Anton Wutz

Academic Editor

PLOS ONE
---

## [Editor Report · Acceptance letter]

4 Nov 2021

PONE-D-21-23013R2 

*Kdm6a* Deficiency Restricted to Mouse Hematopoietic Cells Causes an Age- and Sex-dependent Myelodysplastic Syndrome-Like Phenotype 

Dear Dr. Wartman:

I'm pleased to inform you that your manuscript has been deemed suitable for publication in PLOS ONE. Congratulations! Your manuscript is now with our production department. 

Kind regards, 

on behalf of

Dr. Anton Wutz 

Academic Editor

PLOS ONE